# Dysregulated Host Responses Underlie 2009 Pandemic Influenza-Methicillin Resistant *Staphylococcus aureus* Coinfection Pathogenesis at the Alveolar-Capillary Barrier

**DOI:** 10.3390/cells9112472

**Published:** 2020-11-13

**Authors:** Michaela E. Nickol, Sarah M. Lyle, Brendan Dennehy, Jason Kindrachuk

**Affiliations:** 1Laboratory of Emerging and Re-Emerging Viruses, Department of Medical Microbiology, University of Manitoba, Winnipeg, MB R3E 0J9, Canada; nickolm@myumanitoba.ca (M.E.N.); lyles3@myumanitoba.ca (S.M.L.); dennehyb@myumanitoba.ca (B.D.); 2Vaccine and Infectious Disease Organization-International Vaccine Centre, University of Saskatchewan, Saskatoon, SK S7N 5E3, Canada

**Keywords:** influenza, *Staphylococcus aureus*, alveolar-capillary barrier, coinfection, kinome, virulence factor

## Abstract

Influenza viruses are a continual public health concern resulting in 3–5 million severe infections annually despite intense vaccination campaigns and messaging. Secondary bacterial infections, including *Staphylococcus aureus*, result in increased morbidity and mortality during seasonal epidemics and pandemics. While coinfections can result in deleterious pathologic consequences, including alveolar-capillary barrier disruption, the underlying mechanisms are poorly understood. We have characterized host- and pathogen-centric mechanisms contributing to influenza-bacterial coinfections in a primary cell coculture model of the alveolar-capillary barrier. Using 2009 pandemic influenza (pH1N1) and methicillin-resistant *S. aureus* (MRSA), we demonstrate that coinfection resulted in dysregulated barrier function. Preinfection with pH1N1 resulted in modulation of adhesion- and invasion-associated MRSA virulence factors during lag phase bacterial replication. Host response modulation in coinfected alveolar epithelial cells were primarily related to TLR- and inflammatory response-mediated cell signaling events. While less extensive in cocultured endothelial cells, coinfection resulted in changes to cellular stress response- and TLR-related signaling events. Analysis of cytokine expression suggested that cytokine secretion might play an important role in coinfection pathogenesis. Taken together, we demonstrate that coinfection pathogenesis is related to complex host- and pathogen-mediated events impacting both epithelial and endothelial cell regulation at the alveolar-capillary barrier.

## 1. Introduction

Influenza A viruses (IAVs) infect approximately 10% of the global population each year, resulting in an estimated 3–5 million severe infections and 300,000–650,000 mortalities [1]. This occurs through both seasonal epidemics and sporadic pandemic outbreaks, despite an intensive vaccine program and the existence of antivirals [2,3,4,5]. Infections range from asymptomatic to severe or fatal [6,7] and generally manifest as acute, self-limiting infections in the upper or lower respiratory tract [8,9]. Clinical symptoms include high fever, headache, coryza, cough, myalgias, and general malaise [3,8,9,10]. In healthy adults, symptoms generally peak around 3–5 days post-infection and with convalescence at 7–10 days [8,9,10]. While influenza is generally mild in most of the population, infants <2 years of age, the elderly and individuals with underlying comorbidities (including respiratory, cardiac, neurological or immunosuppressive conditions) are at a high risk of severe disease [8,9,11,12].

There is an increasing appreciation that severe or fatal influenza infections are frequently complicated by bacterial coinfections [13]. The contribution of secondary bacterial infections has been well documented throughout prior influenza pandemics and most notably the 1918 H1N1 influenza pandemic [14]. Modern analyses of lung tissue and review of historical autopsy data from fatal 1918 influenza infections demonstrated that 95% of lethal cases were complicated by a bacterial coinfection, with *Staphylococcus aureus* and *Streptococcus pneumoniae* spp. most commonly identified [15,16,17]. Secondary bacterial pneumonia also resulted in significant morbidity and mortality during both the 1957 and 1968 influenza pandemics, with an estimated 44% of cases being complicated by *S. aureus* and *S. pneumoniae*. More recently, up to 55% of fatal cases during the 2009 H1N1 pandemic were complicated by bacterial coinfections, and methicillin-resistant *S. aureus* (MRSA) was commonly observed in coinfected patients [18,19,20]. Bacterial coinfections also complicate seasonal influenza infections where total influenza-related fatalities are estimated to be 65,000 (including both influenza- and pneumonia-related deaths) in the U.S. annually [18].

Previously, we characterized the host and pathogen molecular mechanisms that contribute to severe influenza-bacterial infections in the lower respiratory tract using a monolayer of an alveolar epithelial cell line [21]. We found that when respiratory epithelial cells were infected with MRSA during peak viral infection, host cell signaling responses shifted from viral- to bacterial-centric as infection moved from the early to late phase [21]. Further, a transition phase in host responses was identified at the mid-point of infection (8–12 h post-MRSA addition), which correlated with a loss of respiratory epithelial barrier function and integrity. While this prior investigation provided important insights into the molecular mechanisms underlying the pathogenesis of influenza-bacterial coinfections, it was limited to epithelial cells alone and did not account for the multicellular complexity of the alveolar-capillary barrier.

To address this, we established a coculture model of the alveolar-capillary barrier by using an in vitro coculture model using primary human alveolar epithelial cells and microvascular endothelial cells. This physiologically relevant model of the lower respiratory tract allowed us to elucidate the molecular mechanisms surrounding barrier dysfunction during IAV-MRSA coinfections. Based on our prior investigations, we hypothesized that secondary bacterial coinfections resulted in severe dysfunction of the alveolar-capillary barrier due to the modulation of bacterial virulence factor expression in the presence of IAV, thus leading to dysregulated host cell signaling responses in both epithelial and endothelial cells at the alveolar-capillary barrier. Our results suggest that the pathogenesis underlying severe influenza-bacterial coinfections in the lower respiratory tract results from both microbial- and host-centric activities.

## 2. Materials and Methods

### 2.1. Virus, Bacteria, and Cell Conditions

The 2009 pandemic H1N1 Influenza A/Mexico/4108/09 (pH1N1; GenBank GQ223112) was kindly provided by Dr. Kevin Coombs (University of Manitoba, Canada). Virus stocks were grown in Madin–Darby canine kidney cells (ATCC, Manassas, VA, USA) maintained in Dulbecco’s modified Eagle medium (Gibco, Grand Island, NY, USA) with 1 µg/mL tosyl phenylalanyl chloromethyl ketone (TPCK)-treated trypsin, concentrated following ultracentrifugation on a 35% sucrose cushion, and kept at −80 °C. Viral titres were determined via plaque assay. CA-MRSA genotype CMRSA10 (USA300; herein referred to as MRSA) was kindly provided by Dr. George Zhanel (University of Manitoba, Winnipeg, MB, Canada). MRSA inocula were generated following growth to the mid-log phase in tryptic soy broth (Hardy Diagnostics, Santa Maria, CA, USA) at 37 °C with shaking. Bacterial titres were determined via the standard plate count. Human (HPAEpiC) were obtained from ScienCell Research Laboratories (Carlsbad, CA, USA). Cells were grown in the airway epithelial basal cell medium fully supplemented with the bronchial epithelial cells growth kit (ATCC, Manassas, VA, USA) at 37 °C and 5% CO_2_. Human pulmonary microvascular endothelial cells (HPMECs) were obtained from ScienCell Research Laboratories (Carlsbad, CA, USA). Cells were grown in the basal endothelial cell medium complete kit (ScienCell Research Laboratories, Carlsbad, CA, USA).

### 2.2. Coculture Model of the Alveolar-Capillary Barrier

The basal side of 0.4 μm transwell inserts (Corning Life Sciences, Montreal, QC, Canada) were coated with the GelTrex LDEV-free reduced growth factor basement membrane matrix (ThermoFisher Scientific, Mississauga, ON, Canada) and rested basal side up for 1 h at 37 °C and 5% CO_2_. Transwell inserts were turned apical side up, the apical side coated with GelTrex, and rested for 1 h at 37 °C and 5% CO_2_. Transwell inserts were turned basal side up, the basal side of the transwell inserts seeded with HPMEC at a concentration of 1.5 × 10^5^ cells/mL (4.5 × 10^4^ cells/cm^2^) in a 1:1 mix of HPAEpiC and HPMEC media, and rested for 3 h at 37 °C and 5% CO_2_. Transwell inserts were turned apical side up and HPAEpiC were seeded on the apical side at a concentration of 3 × 10^5^ cells/mL (9 × 10^4^ cells/cm^2^) in a 1:1 mix of HPAEpiC and HPMEC media. After 24 h, media was removed from the upper compartment of the transwell insert, to allow primary epithelial cells to grow at the air-liquid interface. Media in the lower compartment was refreshed with a 1:1 mix of HPAEpiC and HPMEC media. Cells were permitted to grow to confluency for 14 days, with media in the lower compartment being refreshed every second day. An alveolar-capillary barrier coculture model schematic is presented in Appendix A.

### 2.3. Viral and Bacterial Infection of the Tissue Culture Model

Epithelial and endothelial cells were washed 2× with warm DPBS. Transwell infection media (a 1:1 mix of non-supplemented airway epithelial basal cell medium and basal endothelial cell medium without TPCK-trypsin) was added to the lower compartment and epithelial cells were infected by adding viral inocula to the upper compartment of the transwell insert. Cells were infected with pH1N1 at a multiplicity of infection (MOI) of 0.1 or mock with transwell infection media for 1 h at 37 °C and 5% CO_2_. Following infection, viral inocula were aspirated from cells. Cells were rested for 24 h post-viral infection. Cells were infected with mid-log phase MRSA or mock 24 post-influenza addition with transwell infection media for 1 h. Bacterial MOIs of 0.1 were used and were achieved by serial dilution of mid-log phase culture in transwell infection media as described above. Bacterial inocula were aspirated from cells and both HPAEpiC and HPMEC were harvested at each time point by gentle scraping for further investigation of bacterial replication kinetics, virulence factor modulation, and kinome analysis.

### 2.4. Quantification of Bacterial and Viral Replication Kinetics

Quantification of the total number of adherent and internalized bacteria was determined at 1, 4, 8, 12, 16, 20, and 24 h post-bacterial infection. Respiratory epithelial HPAEpiCs were harvested for bacterial enumeration by washing 2× with DPBS followed by gentle scraping. Cells were pelleted by centrifugation at 5000 rpm for 10 min, supernatant removed, and cells resuspended in 0.025% TritonX-100. Colony forming units (CFU) were quantified by standard bacterial plating on tryptic soy agar (MP Biomedicals, LLC, Solon, OH, USA). Four biological replicates were performed at each time point for enumeration. RT-qPCR was used to quantify viral replication by collecting supernatant samples for IAV–MRSA infected alveolar epithelial cells. Total RNA was extracted from the supernatant using the PureLink Viral RNA/DNA Mini Kit (LifeTechnologies, Burlington, ON, Canada) according to the manufacturer’s instructions. Reverse transcription of total RNA was performed using the Superscript IV first-strand cDNA synthesis kit (Life Technologies, Burlington, ON, Canada) using primers specific for the viral H1N1 HA sequence. Viral genome copy numbers were quantified by comparing RT-qPCR results to an established external viral genome copy number standard.

### 2.5. RNA Extraction, cDNA Synthesis, and Quantitative PCR

Three biological replicates with two technical replicates were collected at 1, 4, 8, 12, 16, 20, and 24 h post-bacterial infection to determine the modification of bacterial virulence factors in the presence of influenza. Following aspiration of media, HPAEpiCs were collected by gentle scraping, pelleting by centrifugation at 1200 rpm for 10 min, and stored at −80 °C until RNA extraction. Standard TRIzol-chloroform extraction (Ambion, Carlsbad, CA, USA) was performed to extract bacterial RNA, before concentration and purity of the RNA were assessed by A_260_:A_280_ spectrophotometry. Total bacterial RNA was normalized to 35 ng and cDNA synthesized using random primers and the QuantiNova reverse transcription kit (Qiagen, Hilden, Germany). Of cDNA 10 ng was amplified in triplicate by RT-qPCR performed on the Applied Biosystems QuantStudio 6 Flex Real-Time PCR system (Life Technologies, Burlington, ON, Canada) using PowerUp SYBR Green Master Mix (Applied Biosystems, Austin, TX, USA) as a detection method and 8 μM of the appropriate primers (Appendix A). Primers were designed and selected using PrimerQuest (https://www.idtdna.com/primerquest). Cycling conditions involved an initial 2 min incubation at 50 °C and a 2 min incubation at 95 °C for SYBR Green activation and polymerase activation, respectively. This was followed by 40 cycles of 15 s at 9 °C for denaturation and 1 min at 60 °C for annealing and extension. Bacterial gene expression was quantified through comparison to the MRSA housekeeping gene 16S, and relative fold change in expression was calculated using the 2^−ΔΔCT^ method [22]. Relative fold change values represent IAV–MRSA (normalized to 16S)/MRSA-alone (normalized to 16S).

### 2.6. Determination of Barrier Integrity in a Coculture Model

The electric cell-substrate impedance sensing trans-epithelial/endothelial electrical resistance (ECIS TEER) 24, 24-well TEER 24 microplates, and common electrode array (Applied Biophysics, Troy, NY, USA) were employed to quantify barrier integrity in a coculture model during pH1N1-MRSA coinfection. Epithelial cells were infected with influenza (MOI 0.1) or mock with infection media (denoted as Time 0) for 1 h followed by resting for 24 h. Viral and mock cells were subsequently infected with mid-log phase MRSA (MOI 0.1) or mock with infection media for 1 h. Resistance measurements were acquired at 4000 Hz every 4 h for 48 h. At each time point, the upper compartment of each transwell insert was filled with 600 μL of infection media and resistance measured for 1 min. Infection media was removed from each transwell insert and the cells allowed to rest until the next time point. Control conditions included: (i) cells infected with influenza-alone (MOI 0.1); and (ii) cells infected with MRSA-alone (MOI 0.1). Three biological replicates were performed per time point and per condition.

### 2.7. Kinome Peptide Array Analysis

Kinome peptide array analysis was performed as previously described [23,24]. IAV-, MRSA-, IAV-MRSA-, and mock HPAEpiCs and HPMECs were collected at 4, 8, 12, and 24 h post-bacterial infection by gentle scraping. Cells were pelleted by centrifugation at 14,000 rpm for 10 min, treated with kinome lysis buffer (20 mM TrisHlC pH 7.5, 150 mM NaCl, 1 mM EDTA, 1 mM EGTA, 1% Triton X-100, 2.5 mM sodium pyrophosphate, and 1× Pierce Halt Protease and Phosphatase Inhibitor) incubated on ice for 10 min, and transferred to fresh microcentrifuge tubes. The Pierce BCA Protein Assay Kit (ThermoFisher Scientific, Mississauga, ON, Canada) was used to quantify total protein concentration. Activation mix (50% glycerol, 50 μM ATP, 60 mM MgCl_2_, 0.05% Brij 35, and 0.25 mg/mL bovine serum albumin) was added to the equivalent amounts of the total protein (100 μg) for each sample, and total sample volumes were matched by the addition of kinome lysis buffer. Kinome peptide arrays (JPT Peptide Technologies GmnbH, Berling, Germany) were spotted with samples and incubated for 2 h at 37 °C and 5% CO_2_. After incubation, arrays were rinsed once with 1% Triton X-100 and once with deionized H_2_O. Arrays were stained using PRO-Q Diamond phosphoprotein stain (Invitrogen, Carlsbad, CA, USA) for 1 h with gentle agitation. Following staining, arrays were washed 3× with kinome destain (20% acetonitrile and 50 mM sodium acetate pH 4.0) for 10 min. Arrays were washed a final time with deionized water for 10 min and dried by centrifugation. A PowerScanner microarray scanner (Tecan, Morrisville, NC, USA) with a 580-nm filter was used to image arrays and Array-Pro Analyzer version 6.3 software (Media Cybernetics, Rockville, MD, USA) was used to collect signal intensity values. Intensity values for spots and background were collected for each array. The Platform for Integrated, Intelligent Kinome Analysis (PIIKA 2) software (available online: https://saphire.usask.ca/saphire/piika) was used to analyze kinome data as previously described [25]. Additional heatmaps were derived using the Heatmapper software suite [26].

### 2.8. Pathway Overrepresentation and Gene Ontology Analysis

Pathway overrepresentation and gene ontology analyses of differentially phosphorylated proteins were performed using InnateDB software as described previously [24,27]. Input data was limited to peptides that demonstrated statistically-significant changes in expression as compared to the respective time-matched mock controls, as described previously [28]. Protein identifiers, phosphorylation fold change values (>1), and *p*-values (<0.05) were uploaded to Innate DB.

### 2.9. Chemokine and Cytokine Measurement

Chemokine and cytokine levels were determined using the microbead array assay Milliplex MAP multiplex kit (Human Cytokine/Chemokine Magnetic Bead Panel 96 Well Plate Assay; Millipore, Billerica, MA, USA) and analyzed on the BioPlex-200 (Biorad, Mississauga, ON, Canada). Supernatants were collected at 4, 8, 12, and 24 h for mock-, pH1N1-, MRSA-, and pH1N1-MRSA-infected samples and stored at −80 °C until use. Supernatants were analyzed according to the manufacturer’s overnight protocol. Lower detection limit was 2.09 pg/mL for EGF, 24.84 pg/mL for FGF-2, 1.73 pg/mL for IFN-α2, 2.02 pg/mL for IFN-γ, 1.84 pg/mL for GRO, 1.66 pg/mL for IL-1β, 1.61 pg/mL for IL-3, 2.47 pg/mL for IL-6, 2.26 pg/mL for IL-8, 1.72 pg/mL for IP-10, 1.95 pg/mL for MCP-1, 1.59 pg/mL for RANTES, 1.55 pg/mL for TNF-α, and 1.53 pg/mL for VEGF.

### 2.10. Statistical Analyses

All numerical data are presented as mean ± SEM. Statistical analyses were performed using ANOVA for comparisons of group means using Prism 8 for MacOS (version 8.2.1). This includes pathogen replication kinetics, RT-qPCR, ECIS, and Milliplex MAP multiplex kit. A *p* value of ≤ 0.05 was considered statistically significant for all analyses. *p* values less than 0.05 are summarized by a single asterisk (*), less than 0.01 are summarized by two asterisks (**), less than 0.001 are summarized by three asterisks (***), and less than 0.0001 are summarized by four asterisks (****).

## 3. Results

### 3.1. MRSA Replication Kinetics Are Similar during MRSA-Alone and pH1N1-MRSA Infection

We first sought to determine how pre-existing pH1N1 infection affects bacterial replication in an in vitro tissue culture model of the alveolar-capillary barrier. Primary HPMECs were seeded on the basal side of transwell inserts and HPAEpiCs were seeded on the apical side. Temporal enumeration of bacteria was investigated by adding MRSA to our mock or pH1N1-infected tissue culture model 24 h post-infection. The number of adherent and internalized bacteria in epithelial cells was quantified through standard bacterial plating (Figure 1). No bacteria were identified by plating from the endothelial cells.

While there appeared to be a trend towards faster bacterial replication in pH1N1-MRSA coinfection, no statistically significant differences were observed between either infection condition (*p* = 0.3258) or between either infection condition over time (*p* > 0.6000). This suggested that MRSA fitness within pulmonary respiratory epithelial cells at the alveolar-capillary barrier is not affected by the presence of pH1N1. In contrast, viral loads decreased over time post-MRSA infection (Appendix A). These matched our previous observations [21]. No virus was identified by RT-qPCR in the endothelial cells.

### 3.2. Modulation of Bacterial Virulence Factors in the Presence of pH1N1

As our results suggested that coinfection did not result in altered bacterial replication kinetics, we next sought to characterize how the modulation of bacterial virulence factors related to adhesion and invasion might be altered during coinfection. Our prior work with coinfection in A549 cells demonstrated that altered virulence factor expression was only found during early infection (1–4 h post-MRSA addition). Thus, here we focused on the same time points. The alveolar pulmonary cells of our coculture model were infected with pH1N1 at a MOI of 0.1 or mock and allowed to rest for 24 h prior to MRSA-infection (MOI 0.1). Cell lysates were collected at multiple time points post-infection and RT-qPCR employed to examine differential modulation of MRSA virulence factor gene expression in the presence or absence of pre-existing influenza virus infection. We studied 13 virulence factor genes directly related to adhesion and invasion: coa, ebpS, eno, fnbA, fnbB, hla, hlgA, icaA, icaB, sbi, sek, seq, and spA. Modulation of virulence factors was observed at 1 and 4 h post-MRSA infection, which mimicked our prior results in immortalized A549 cells [21].

At 1 h post-infection, *eno* (*p* = 0.0120), *icaB* (*p* < 0.001), *sek* (*p* = 0.0146), and *seq* (*p* = 0.0135) were significantly upregulated at 1 h in coinfected samples as compared to MRSA-alone (Figure 2). At 4 h post-MRSA infection, *coa* (*p* < 0.0001), *fnbB* (*p* < 0.0001), *hla* (*p* = 0.0014), *hlgA* (*p* < 0.0001), *icaA* (*p* < 0.0001), *icaB* (*p* < 0.0001), *sbi* (*p* < 0.0001), and *sek* (*p* < 0.0001) were significantly upregulated during coinfection. This data coincides with the lag phase of MRSA in the presence of pH1N1 at 1 and 4 h, suggesting that adhesion- and invasion-associated virulence factors may play a role in the initial stages of MRSA infection in primary alveolar cells previously infected with pH1N1.

### 3.3. Barrier Integrity of a Coculture Model of the Alveolar-Capillary Barrier during pH1N1-MRSA Coinfection

We next sought to characterize the effect of pH1N1-MRSA coinfection on barrier integrity in our coculture model by measuring temporal changes in resistance. Cells were either mock or infected with pH1N1 at a MOI of 0.1 (first arrow; designated as Time 0), allowed to rest for 24 h, and either mock or infected with MRSA at a MOI of 0.1 (second arrow). No change in resistance was observed following pH1N1-alone infection as compared with mock cells; the resistance of each of the observed conditions remained steadily at 110 ohms (Figure 3). Following bacterial addition, infection with MRSA-alone resulted in no changes in resistance. No significant differences in barrier integrity were observed at any time point between models infected with MRSA-alone and pH1N1-alone. Samples coinfected with pH1N1-MRSA resulted in a steady decrease in resistance beginning at 8 h post-MRSA addition (30 h). By 45 h, pH1N1-MRSA was significantly downregulated (*p* = 0.0005) as compared with the mock model. This decrease in barrier resistance beginning at 8 h post-MRSA infection coincided with the beginning of the exponential phase of MRSA in the presence of pH1N1.

### 3.4. Temporal Analysis of the Host Kinome Response in a Coculture Model of the Alveolar-Capillary Barrier during pH1N1-MRSA Coinfection

As our temporal analysis of barrier integrity suggested that pH1N1-MRSA coinfection results in more severe barrier dysregulation compared with either pathogen alone, we addressed whether aberrant cell-mediated immune responses contribute to coinfection pathogenesis. We performed temporal kinome analysis of pH1N1-, MRSA-, and pH1N1-MRSA-infected alveolar epithelial and microvascular endothelial cells of the alveolar-capillary cocultures. Time-matched mock controls cells served as controls. Alveolar-capillary barrier cocultures were initially infected with pH1N1 (MOI 0.1) or mock and rested for 24 h prior to bacterial infection. MRSA addition to cells (+ or – pH1N1) was designated as time 0. Epithelial and endothelial cells were harvested separately at 4, 8, 12, and 24 h post-MRSA infection. Both pH1N1-alone infected cells and mock control cells were treated with MRSA-free infection inoculum at time 0 to normalize cellular responses resulting from physical stress during inoculum addition. Time-matched pH1N1-, MRSA-, and mock control cells were collected throughout the duration of the experiment.

To gain insight into the host kinome response of pulmonary epithelial cells during pH1N1-MRSA coinfection as compared to infection with either pathogen alone, biological subtraction of the time-matched mock kinome datasets from their respective infected counterparts was performed. Respective hierarchical clustering analysis of the kinome data following mock background subtraction is presented in Figure 4. Notably, each of the time-matched samples from IAV-alone, MRSA-alone, and IAV–MRSA infected samples clustered together, resulting in four major clusters. From left to right, the first cluster (denoted as A) consisted of each of the 4 h time-matched samples, the second cluster (denoted as B) consisted of each of the 12 h time-matched samples, the third cluster (denoted as C) consisted of each of the 8 h time-matched samples, and the fourth cluster (denoted as D) consisted of each of the 24 h time-matched samples. Clusters B and C, consisting of the 8 and 12 h time points, respectively, clustered together more strongly than with the samples from 4 and 24 h post-MRSA infection. Moreover, the 24 h samples differentiated most strongly from each of the other time points. This data suggested that the modulation of the host kinome response were strongly related to post-infection time points, with intra-time point dependent differences in host responses to infection.

We next sought to identify host cell signaling responses or biological networks in the pH1N1-MRSA-infected pulmonary alveolar cells that were selectively modulated at 24 h post-MRSA infection. Kinome analysis at 24 h post-MRSA addition demonstrated that pH1N1-MRSA coinfection resulted in the activation of numerous signaling pathways as compared with either pH1N1- or MRSA-infection alone (Appendix A). There was an overrepresentation of numerous pathways related directly to the cell cycle, TLR-related signaling, interleukins, and interferon signaling. TLR pathways were not identified in pH1N1- or MRSA-alone datasets, while there was a unique over-representation of TLR signaling pathways and TLR-associated pathways during coinfection. Infection with pH1N1-alone resulted in a lower total number of pathways identified as compared to the coinfection data with the overrepresentation of IFN signaling pathways (lowest *p*-value), cytokine signaling, and apoptosis-associated pathways (Appendix A). This suggests that IFN-mediated responses are muted during coinfection as compared to pH1N1 alone in alveolar epithelial cells. In contrast, MRSA infection alone resulted in relatively few upregulated pathways as compared with the mock control, namely IGF- and inflammasome-associated signaling pathways (Appendix A). This data demonstrates that pH1N1-MRSA coinfection results in a unique cell response signature in primary differentiated alveolar epithelial cells grown in close proximity to pulmonary endothelial cells. This contrasts to our prior analysis of coinfection in A549 cells where kinome responses from pH1N1- and MRSA-alone infections largely overlapped.

Pulmonary endothelial cells from the cocultures were also isolated throughout the time course of infection. Biological subtraction of time-matched mock kinome datasets was performed as previous (Figure 5). Similar to the epithelial cells, biological subtraction revealed four major clusters matched by time point. From left to right, the 8 h time points of pH1N1-alone, MRSA-alone, and pH1N1-MRSA infected samples clustered together (denoted as A). The second cluster consisted of pH1N1-alone, MRSA-alone, and pH1N1-MRSA infected samples at 12 h post-bacterial infection (denoted as B). The third cluster was comprised of the 24 h time points of IAV-alone, MRSA-alone, and IAV–MRSA infected samples (denoted as C). Lastly, samples infected with IAV-alone, MRSA-alone, and IAV–MRSA at 4 h post-bacterial infection clustered together (denoted as D). A and B clustered together; C and D formed a separate cluster. This suggested that the modulation of the host kinome response related strongly to time post-infection as also seen in the epithelial cells.

Coinfection resulted in the overrepresentation of signaling pathways related to hedgehog-, proteasome-, cell stress-, and Wnt- -related pathways at 24 h (Appendix A). In contrast, only a single signaling pathway was differentially modulated in the endothelial cells during pH1N1-alone infection, which was unsurprising given the lack of susceptibility of these cells to pH1N1 transmission from the virus-infected epithelial cells. MRSA-alone infection showed notable upregulation of hedgehog- and Wnt/β-catenin-associated signaling pathways as compared with time-matched mock controls (Appendix A). Similar to the coinfected cells, MRSA infection alone resulted in over-representation of signaling pathways related to hedgehog signaling and Wnt/β-catenin-related signaling suggesting that endothelial cell host responses are largely driven by responses to the MRSA-infected epithelial cells during coinfection (Appendix A).

### 3.5. Cytokine Expression Is Modulated during pH1N1-MRSA Coinfection

As our kinome data revealed that pathways related to interleukins and cytokine signaling were both overrepresented when alveolar epithelial respiratory cells were infected with pH1N1 and pH1N1-MRSA, we next sought to characterize the expression of proinflammatory cytokines by pH1N1-, MRSA-, and pH1N1-MRSA infected cells. Supernatants were collected at 4, 8, 12, and 24 h post-MRSA addition and cytokine secretion was assessed for each condition. Cytokines that were measured included the epidermal growth factor (EGF), fibroblast growth factor 2 (FGF-2), IL-6, IL-8, interferon-γ induced protein 10 (IP-10/CXCL10), monocyte chemoattractant protein-1 (MCP-1/CCL2), and VEGF. Significant downregulation of EGF expression was observed at 12 h in both pH1N1-alone (*p* = 0.0003) and pH1N1-MRSA (*p* = 0.0002) infection and was resolved by 24 h (Figure 6).

At 4 h, FGF-2 expression was significantly upregulated in pH1N1-alone infection (*p* = 0.0052) and also at 12 h and 24 h. FGF-2 was significantly upregulated in the case of pH1N1-alone (*p* = 0.0035) and pH1N1-MRSA coinfection (*p* = 0.0140) at 12 h, and in the case of pH1N1-alone (*p* = 0.0008) and pH1N1-MRSA (*p* = 0.0002) at 24 h. MRSA-alone infection resulted in significant repression at 12 h compared with mock-infection (*p* = 0.0001). Significant upregulation of IL-6 was observed at 24 h (*p* < 0.0001) in pH1N1-alone infections, and at 8 and 24 h (*p* = 0.0303 and *p* < 0.0001, respectively) in pH1N1-MRSA coinfections. IL-8 was significantly upregulated at 4, 8, 12, and 24 h in pH1N1-alone infection (*p* = 0.0059, *p* = 0.0056, *p* = 0.0280, and *p* < 0.0001, respectively). IL-8 was also significantly upregulated at 4, 8, and 24 h in pH1N1-MRSA coinfection (*p* = 0.0002, *p* < 0.0001, and *p* < 0.0001). In MRSA-alone infections, IL-8 was significantly upregulated at 24 h only (*p* = 0.0016).

IP-10 was significantly upregulated at 4, 8, 12, and 24 h in pH1N1-alone infection and pH1N1-MRSA coinfection (*p* < 0.0001 for each), suggesting that the presence of pH1N1 has an important effect on IP-10 secretion. MCP-1 was only significantly upregulated at 24 h in pH1N1-alone infection (*p* < 0.0001), but not at any other time point. VEGF was significantly repressed at 4 h in each infection condition as compared with the mock samples (*p* < 0.0001 for all) and remained repressed at the 8 h timepoint in the coinfection samples alone (*p* = 0.0019). Taken together, this data suggest that cytokine expression during coinfection was largely driven by pH1N1 infection and may play an important role in barrier disruption.

## 4. Discussion

As severe influenza and influenza-bacterial coinfections within the lower respiratory tract can lead to disruption of the alveolar-capillary barrier with potentially deleterious effects on both gas exchange and normal lung function, we sought to examine influenza-bacterial coinfections in a physiologically relevant model of the alveolar-capillary barrier. No significant differences were found for MRSA replication kinetics at any time point across each of our infection conditions similar to our prior analysis in A549 cells [21]. This suggests that MRSA fitness is not altered by pre-existing pH1N1 infection at the alveolar-capillary barrier. Further, this suggests that the increased disease severity associated with influenza-bacterial coinfection is not simply due to increased bacterial burden within the lungs during coinfection. The expression of MRSA virulence factors related to adhesion and invasion were selectively upregulated at 1 and 4 h in our coculture model, which was similar to our prior findings [21]. The pattern of upregulated gene expression corresponded with the lag phase of MRSA at 1 and 4 h in the presence of pH1N1. At 1 h post-MRSA infection, significant upregulation was only observed in *eno*, *icaB*, *sek*, and *seq*. The *eno* protein codes for enolase, which binds to laminin in the basal lamina of the epithelia; previous studies have shown that the high prevalence of *eno* could play an important role in future MRSA vaccine design, which is further underlined by our results [29,30]. The ica locus, which codes for *icaB*, is involved in intracellular adhesion and biofilm formation [31,32,33,34,35]. Upregulation of *icaB* early in infection may be indicative of the increased lag phase of MRSA in the presence of pH1N1 during alveolar epithelial infection. This upregulation of *icaB* was not observed in our alveolar monolayer and may have been due to the production of surfactant by our primary differentiated alveolar cells [21]. Lastly, *seq* and *sek* encode secreted exotoxins that alter the host cell membranes, resulting in lysis [36,37,38]. The superantigen properties of *sek* and *seq* also contribute directly to MRSA virulence [39,40,41]. Lysis of neutrophils by exotoxins such as *sek* and *seq* result in reactive oxygen species release, which leads to damage and inflammation to surrounding lung tissue [39,41]. Further, mouse models have suggested that the *sek* and *seq* superantigens of MRSA play a role in T-cell signaling responsible for much of the early lung damage seen in *S. aureus* infection [40]. At 4 h post-MRSA addition, significant upregulation seen in *coa*, *fnbB*, *hla*, *hlgA*, *icaA*, *sbi*, and *sek*. The product of *coa*, coagulase, plays a role in initial adhesion to epithelial cells by cleaving fibrinogen and activating prothrombin [42,43,44]. The fnb locus, which codes for *fnbB*, is also involved in initial bacterial adhesion to epithelial cells for internalization [38,43,45,46,47,48]. As both *hla* and *hlgA* result in the lysis of infected cells, this may support the idea that bacterial toxins play a role in secondary bacterial pathogenesis early in infection of alveolar epithelial cells, as both virulence factors were also upregulated during coinfection of our alveolar monolayer [21]. Upregulation of *icaA*, which plays a similar role to *icaB*, may underlie the increased lag phase observed in pH1N1-MRSA coinfection [31,33,34,35]. Lastly, the upregulation of *sbi* may suggest that pre-existing pH1N1 infection in alveolar epithelial cells may indirectly facilitate a strong immune evasion response in MRSA through interaction of bacteria with secreted messengers or agonists from damaged epithelium resulting in the inhibition of antibody responses through the binding of IgG and C3, a novel immune evasion approach [49].

Coinfection had significant effects on alveolar-capillary barrier permeability as assessed by ECIS. As the presence of MRSA alone did not result in permeabilization, our data suggest that underlying pH1N1 infections at the alveolar-capillary barrier contribute to dysfunction following secondary bacterial coinfections. This is unsurprising as pH1N1-alone infection in healthy adults rarely results in severe disease [11,12]. Likewise, severe illness and death in otherwise healthy adults infected with *S. aureus* is often associated with prior influenza infection [50,51,52]. We also utilized kinome analysis to provide insights into host response modulation during pH1N1-MRSA coinfections. Our human kinome peptide arrays had an increased breadth of kinase recognition sequences as compared to our prior analysis in A549 cells (1294 targets compared to 309 targets) [21]. The use of a transwell coculture system provides for intercellular interactions between the epithelial and endothelial cells. Interestingly, in our primary epithelial cells, the 8 and 12 h time points still clustered together more strongly than with the 4 and 24 h time points.

Our pathway over-representation analysis of differentiated alveolar epithelial cells further supports our postulate that IAV-bacterial coinfections are able to specifically modulate host cellular responses independently of infection with pH1N1- or MRSA-alone. Most notably, pathways related to TLRs, including TLR-2, -4, -5, -7/8, -9, and -10 were overrepresented only in IAV–MRSA coinfected cells. TLRs are an important aspect of severe IAV- MRSA coinfections [53,54]. MyD88, an important mediator of TLR signaling and NK-κB activation, was also strongly overrepresented in our coinfected epithelial cells [55,56,57]. Interestingly, *S. aureus* can dampen TLR-2 activation and thus NK-κB activation [57]. Taken together, our data supports the hypothesis that further characterization of TLR signaling in pH1N1-MRSA coinfection may provide new opportunities for targeted drug therapies. Cell cycle pathways were also overrepresented in coinfected epithelial cells. Many of these overrepresented pathways were involved in the G1 and G2 phases, and the transition between these two phases. Previous gene-expression analysis in patients infected with pH1N1 has shown that progression towards severe IAV infection is often characterized by abnormal deviations in cell cycle and apoptosis signaling pathways [58]. Specifically, progression to severe infection was characterized by increased aberrant DNA replication in the G1/S phase but delayed exit from the G2/M phase. Cytokine signaling was overrepresented at 24 h in the presence of pH1N1, whether MRSA was present or not. “Cytokine Signaling in Immune System” was highly upregulated (56 associated proteins from our kinome arrays) in both pH1N1-MRSA and MRSA-alone infection conditions in alveolar epithelial cells. Infection with pH1N1-alone also resulted in upregulation in “Signaling by Interleukins”, with 44 associated proteins from our kinome arrays, in addition to upregulation of IL-1, IL-2, and IL-6. Various studies have suggested that severe lung inflammation seen in highly pathogenic influenza strains may be due to increased levels of proinflammatory cytokines [58,59,60,61].

In comparison to pH1N1-MRSA and pH1N1-alone infection, relatively few pathways were upregulated in alveolar epithelial cells infected with MRSA-alone. Notably, upregulation of pathways related to hedgehog, Wnt, and β-catenin signaling was observed. Our analysis demonstrated that coinfection resulted in a paucity of upregulated signaling pathways compared to the epithelial cells. Overrepresentation of pathways was most commonly found in pH1N1-MRSA infection, followed by MRSA-alone infection, and pH1N1-alone infection. Only one pathway was overrepresented in cells infected with pH1N1-alone. This was perhaps unsurprising as no evidence of pH1N1 infection was identified in the endothelial cells although the alveolar epithelial cells were susceptible to productive viral infection. Pathways related to hedgehog and Wnt/β-catenin signaling were most commonly overrepresented in endothelial cells in both our MRSA-alone and pH1N1-MRSA infection conditions. The Wnt signaling pathway regulates a number of genes involved in cell growth, differentiation, survival, and immune functions [62]. However, recent studies have revealed both a pro- and anti-inflammatory response from Wnt/β-catenin signaling, suggesting that the role of Wnt may be dependent on the stimulus, cell type, and crosstalk with other signaling pathways [62,63,64,65,66]. The majority of evidence notes that activation of the Wnt pathway is able to reduce inflammatory processes triggered by bacterial pathogens [62]. As MRSA is not present in the endothelial cells of the coculture model, overrepresentation of pathways related to Wnt signaling may be indicative of the ability of the alveolar epithelial cells to communicate with the underlying endothelial cells of the capillary.

Based on our kinome analysis we also assessed the concentration of cytokines that were previously implicated in severe pH1N1-MRSA infections. The proinflammatory response may be an important factor in disease outcome from secondary bacterial pneumonia, as various studies have suggested that severe lung inflammation in influenza infections may be related to increased levels of proinflammatory cytokines in the lung [58,59,60,61]. Specifically, pH1N1 is known to induce the expression of a number of interleukins in both the respiratory tract and central nervous system [67]. Significant downregulation of EGF was observed at 12 h in both pH1N1-alone and pH1N1-MRSA infection. EGF is a growth factor capable of stimulating proliferation of epithelial cells by activating cellular signaling through engagement of the EGF receptor [68]. A previous study reported that EGF was significantly higher in healthy patients compared to pH1N1 infection and the authors suggested that EGF was actively suppressed, in an effort to protect the lung from host or virus mediated damage [69]. This may explain the significant downregulation of EGF observed in our pH1N1-alone and pH1N1-MRSA infected cells. Significant upregulation of FGF-2, a member of the fibroblast growth factor family, was also observed in infection with pH1N1-alone and in pH1N1-MRSA coinfections. Conversely, FGF-2 was significantly downregulated at 8 h in cells infected with MRSA-alone. FGF-2 plays an important role in epithelial repair in the lung and in wound healing [70,71,72]. FGF-2 dysregulation is implicated in many inflammatory diseases, and a study in mice suggested that FGF-2 plays a vital role in IAV-induced lung injury [73]. While our data seems to suggest that pH1N1 infection results in a significant increase in FGF-2 expression regardless of the presence of MRSA, further investigation will need to be done to fully understand the role of FGF-2 in severe pH1N1-MRSA infections. IL-6 was significantly upregulated at multiple time points in pH1N1-MRSA infections, and at 24 h in pH1N1-alone infections. IL-6 levels have been shown to be significantly elevated in the presence of a clinically relevant secondary bacterial infection, which may make its upregulation in pH1N1-MRSA coinfection unsurprising [54,74]. It is enticing to speculate that IL-6 upregulation may be related to increased barrier permeability found in our pH1N1-MRSA ECIS results, as high levels of IL-6 are known to directly damage endothelial cells [75]. Further, elevated serum IL-6 levels have been implicated as a potential biomarker for disease severity in pH1N1-alone infections [76]. IL-8 was also significantly upregulated in pH1N1-alone and pH1N1-MRSA infections. IL-8 shows distinct target specificity for attracting and activating neutrophils to inflammatory regions [77]. IP-10 was significantly upregulated at all time points in pH1N1-MRSA and pH1N1-alone infections. This is perhaps unsurprising as previous studies have reported that interferon-related signaling, such as IP-10, were more abundant in cases of severe disease [74]. IP-10 is able to directly influence apoptosis in disease, which may explain the upregulation of apoptosis pathways in our epithelial cell kinome data from pH1N1-MRSA and pH1N1-alone infections [78]. Interestingly, IP-10 was decreased in expression in the coinfected samples as compared to pH1N1 alone. Prior analysis has demonstrated that *S. aureus* downregulates IP-10 production [79] and this difference likely represents the inhibitory activity imparted by MRSA during coinfection. Similarly, MCP-1 was significantly upregulated in pH1N1-infected cells at 24 h but not in MRSA-alone or coinfected cells. This is perhaps unsurprising as MCP-1 was not upregulated in response to *S. aureus* infection in airway epithelial cells [80] and may reduce the overall induction of this chemokine during coinfection. Lastly, VEGF was significantly downregulated across all conditions early during infection as compared to mock cells. VEGF is a regulator of cell growth, and is most abundant in the lung; transcripts are primarily localized in alveolar type II cells in the alveoli [81,82,83]. The downregulation observed in our model is perhaps surprising, as previous studies have reported that hypoxia, commonly seen in severe pH1N1-MRSA infection, results in increased induction of VEGF from ATII cells [81]. Further investigations in more advanced in vitro or in vivo models of severe pH1N1-MRSA and/or a longer course of infection may better describe the role VEGF plays at the alveolar-capillary barrier.

Overall, significant upregulation of cytokine expression is often observed in pH1N1-MRSA coinfection. Further understanding of how elevation of specific cytokines impact pH1N1-MRSA disease severity may reveal potential additional therapeutic targets to reduce the generation of a cytokine storm in infected patients. This strategy has previously shown promise as inhibition of certain cytokines, such as IL-10, have shown improved survival from bacterial pneumonia late after influenza infection [84,85].

## 5. Conclusions

Our investigations into influenza–bacterial coinfections in a primary coculture model of the alveolar-capillary barrier suggest that infection with both pH1N1 and MRSA appears to have a synergistic pathologic effect, as infection with either pathogen on its own did not result in the loss of barrier integrity nor strong dysregulation of host response during coinfection. Strikingly, dysregulation of the host response seems to be driven primarily by the response of alveolar epithelial cells to both pathogens. This is contrary to what was observed in an alveolar cell line, which suggested that MRSA-alone infection resulted in a similar dysregulation as pH1N1-MRSA coinfection [21]. This disparity could be due to a number of reasons. Notably, our results indicate that alveolar-capillary barrier cocultures are likely a more reliable surrogate for assessing lower respiratory tract pathogenesis than alveolar epithelial cells alone, though further investigations are needed to confirm this. Additionally, the coculture model is able to reflect the crosstalk that is able to occur between the alveolar epithelial and microvascular endothelial cells at the alveolar-epithelial barrier.

## Figures and Tables

**Figure 1 cells-09-02472-f001:**
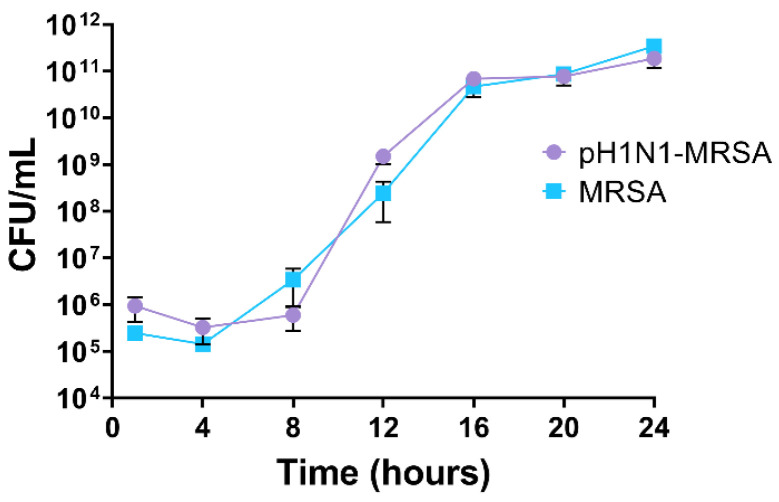
MRSA replication kinetics during MRSA infection and pH1N1-MRSA coinfection in primary alveolar epithelial cells. Human primary epithelial cells of the alveolar-capillary barrier were infected with pH1N1 (MOI 0.1) or mock followed by MRSA infection 24 h later (MOI 0.1). CFU were quantified by standard bacterial plating. Error bars represent SEM calculated from three biological replicates (*n* = 3). Statistical analyses were performed using ANOVA for comparisons of group means using Prism 8.

**Figure 2 cells-09-02472-f002:**
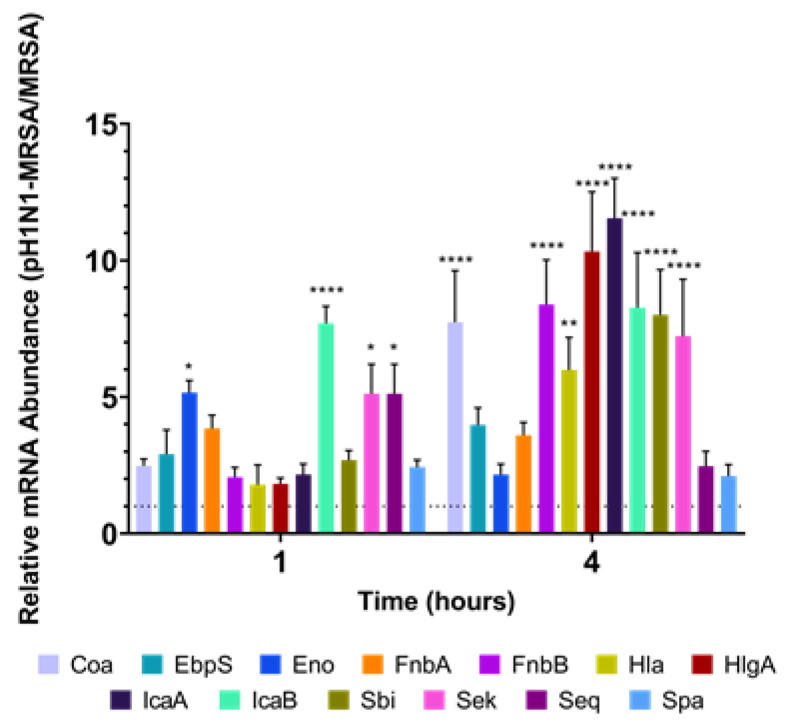
MRSA virulence factors are modulated during coinfection at the alveolar-capillary barrier. RT-qPCR was employed to examine differential modulation of relative MRSA virulence factor mRNA abundance at 1 and 4 h in infected primary epithelial cells of the alveolar-capillary barrier. Relative mRNA abundance fold changes represent pH1N1-MRSA vs. MRSA infection alone and were calculated by the 2^−ΔΔCT^ method. The dashed line signifies a fold-change of 1. Error bars represent SEM calculated from three biological replicates (*n* = 3). Statistical analyses were performed using ANOVA for comparisons of group means using Prism 8. *: *p* < 0.1 **: *p* < 0.01 ****: *p* < 0.0001.

**Figure 3 cells-09-02472-f003:**
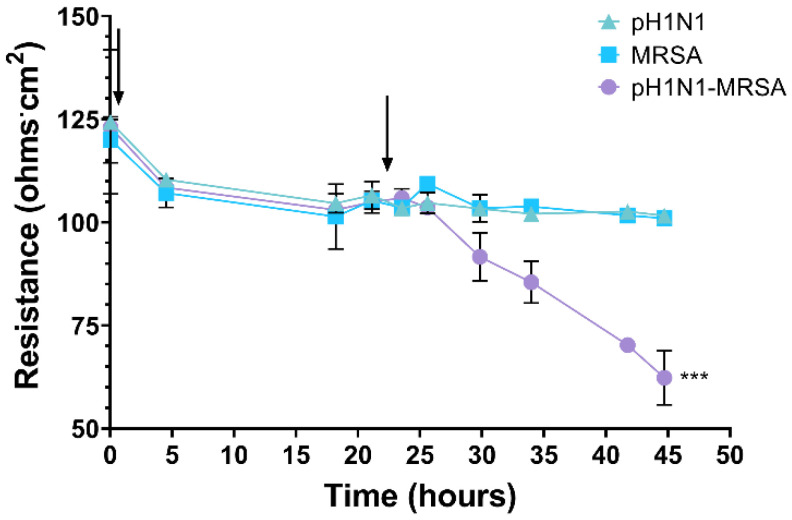
pH1N1-MRSA coinfection decreases barrier function in an alveolar-capillary coculture model. Human primary epithelial cells of the alveolar-capillary barrier were infected or mock with pH1N1 (MOI 0.1; first arrow) and MRSA (MOI 0.1; second arrow) was added to cells 24 h later. Error bars represent SEM calculated from three biological replicates (*n* = 3). Statistical analyses were performed using ANOVA for comparisons of group means using Prism 8. ***: *p* < 0.001.

**Figure 4 cells-09-02472-f004:**
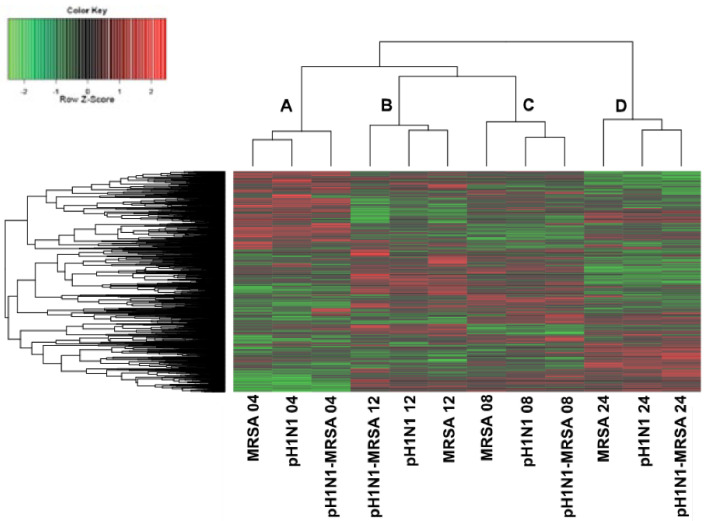
Temporal kinome responses of pH1N1, MRSA, and pH1N1-MRSA infection in epithelial cells within an alveolar-capillary barrier coculture model. Mock kinome responses were subtracted from time-matched infected samples. Fold change phosphorylation values are plotted for all kinase recognition sequences. Red depicts upregulation, while green depicts downregulation as compared with the background. A–D designate the four major dataset clusters as identified following hierarchical clustering.

**Figure 5 cells-09-02472-f005:**
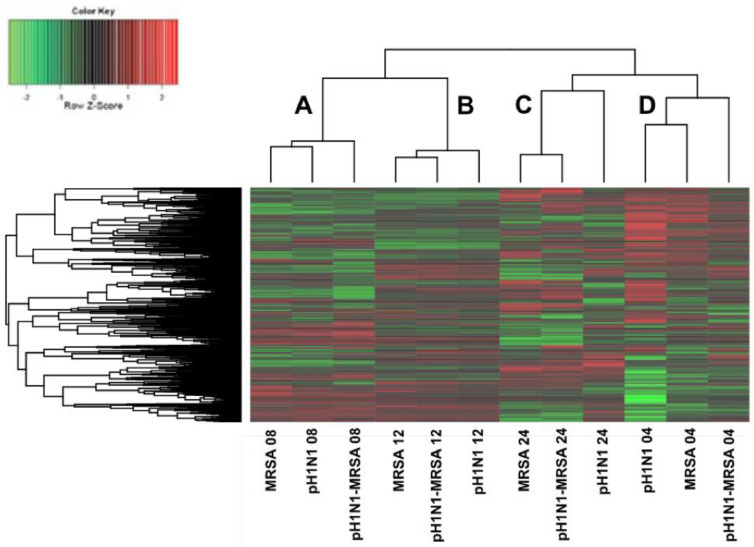
Temporal kinome responses of pH1N1, MRSA, and pH1N1-MRSA infection in pulmonary endothelial cells within an alveolar-capillary barrier coculture model. Mock kinome responses were subtracted from time-matched infected samples. Red depicts upregulation, while green depicts downregulation as compared with the background. A–D designate the four major dataset clusters as identified following hierarchical clustering.

**Figure 6 cells-09-02472-f006:**
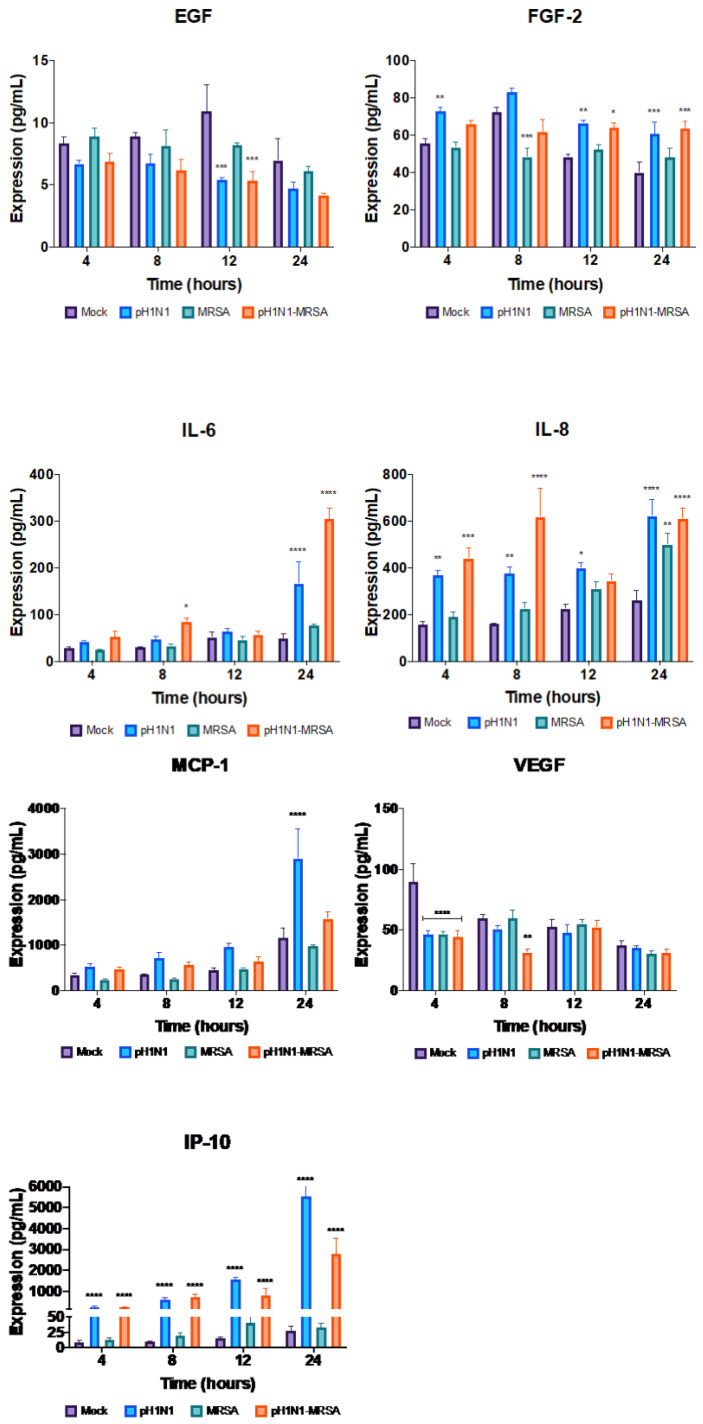
Induction of cytokine expression during pH1N1-, MRSA-, and pH1N1-MRSA infection in the alveolar-capillary barrier model. Primary epithelial cells of the alveolar-capillary barrier were mock-infected or infected with pH1N1 (MOI 0.1). MRSA (MOI 0.1) was added 24 h later. Cytokine levels were determined using the Milliplex MAP multiplex kit. Error bars represent SEM calculated from three biological replicates (*n* = 3). Statistical analyses were performed using ANOVA for comparisons of group means using Prism 8. *: *p* < 0.1 **: *p* < 0.01 ***: *p* < 0.001 ****: *p* < 0.0001.

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
