# Peer review of "Dysregulated Host Responses Underlie 2009 Pandemic Influenza-Methicillin Resistant Staphylococcus aureus Coinfection Pathogenesis at the Alveolar-Capillary Barrier"

_cells, 2020, doi:10.3390/cells9112472_

Round 1
Reviewer 1 Report
The manuscript by Nickol et al describes investigations about a very important topic of bacterial and viral co-infections of the respiratory tract. The authors performed a series of experiments with influenza A virus and S. aureus. They utilized a complex co-culture model of alveolar-capillary barrier for the infection experiments. The authors conclude that TLR- and inflammatory cell signalling and cytokine secretion play an important role in co-infections.
The entire study design sounds very plausible. However, I have to admit that I disagree with many conclusions provided by the authors. Specifically, the authors tend to overinterpret and "over"discuss the results. Furthermore, I personally think that the manuscript is way too long and should be shortened. At the same time, more informations need to be provided in the figure legend and table sections.
Specific comments:
- Line 85: which specific MRSA USA300 was used in this study? There are many USA300 strains.
- Lines 212-225: Already the justification why this experiment was performed is way too long (lines 213-222). The methodology was explained on four additional lines (222-225).
- Although mentioned in the method section (lines 128-129), data on viral replication on both cell types or supernatants is not provided by the authors.
- Lines 226-253: According to the method section, both cell types were harvested and bacterial numbers were determined (lines 118-120), however, the authors provide data on epithelial cells only. What is the rationale behind that?
- Lines 226-253: The author describe nicely what is shown in Fig. 1. However, if there are no significant differences, there is also no need to convince the reader of trends which are not present. The athors conclude correctly (lines 251-252) that there is no impact of H1N1 infection on bacterial replication. Therefore, some of the section can be easily deleted from this results parts. Unfortunately, this kind of arguing continues through the entire result section of the manuscript and should be improved by the authors.
- Figure 1. Is this a representative graph? How many experiments were performed? There are no SDs or SEMs.
- Lines 263-265: The authors state that modulation of analyzed bacterial genes was observed only at 1 and 4 h post infection. Figure 2 depicts only these two time points. No data on other time points is provided.
- Lines 266-267 and lines 270-271: The authors state: each of the 13 genes were upregulated. Again, this is not true. If no significant differences are calculated, consequently no upregulation of the gene is detected.
- Figure 2: Which statistical test was used?
- Figure 3: Is this a representative graph? How many experiments were performed? There are no SDs or SEMs.
- Figure 3 and lines 281-303: the authors describe significant differences, however, the values or "stars" are not provided in the corresponding figure.
- Lines 281-303: the sentense: "at 90 h, pH1N1-MRSA..." does not make any sence. The data of 90 h is not provided.
- I really like the fact that the authors distinguish between the epithelial and capillary compartment in the kinome analyses. However, I mostly disagree with the presented conclusions. To my knowledge, pathway analyses are ranked based on p-values. I addition, the number of significantly differentially expressed proteins out of the total number of proteins which are assigned to a particular pathway are important for such calculations. I will give just one example how I would interprete the data based on Table S2. I would add another row to each table showing how many proteins are assigned to each calculated pathway. Furthermore, I would remove all pathways which have a p-value =0.05 or higher. Indeed, cytokine signalling and signalling by interleukines would remain. But it is not one of the top-ten pathways (based on p-values). I would only be able to claim it, if 56 kinases out of e.g. 200 (totaly assigned) would be covered (=28%) while estrogen signalling would have e.g. 4 out of 200. Just by looking at the data of epithelial cells (Tables S2-S4), I would conclude that e.g. Interferon signalling plays a role in viral infections (Table S3 (last significant pathway);typical viral response) which is dampened by bacterial infection of H1N1 infected cells (Figure S2; p-value>0.05). In bacterial infections itself, this signalling does not play a role (no IFN signaling at all). This is how I would approach it, of course by using the whole data set.
- Similar commets apply to the cytokine data (Figure 6) and the corresponding result section. Please comment only significat differences. Trends are not differences!
- Based on the results, I would conclude that EGF does not play a role. In the remaining cytokine analyses, only viral signature is present in co-infections. No bacterial imprint is seen.
- Discussion: this part is way too long and needs to be shortened and adjusted to the improved results version.
Author Response
The manuscript by Nickol et al describes investigations about a very important topic of bacterial and viral co-infections of the respiratory tract. The authors performed a series of experiments with influenza A virus and S. aureus. They utilized a complex co-culture model of alveolar-capillary barrier for the infection experiments. The authors conclude that TLR- and inflammatory cell signalling and cytokine secretion play an important role in co-infections.
The entire study design sounds very plausible. However, I have to admit that I disagree with many conclusions provided by the authors. Specifically, the authors tend to overinterpret and "over"discuss the results. Furthermore, I personally think that the manuscript is way too long and should be shortened. At the same time, more informations need to be provided in the figure legend and table sections.
Specific comments:
Line 85: which specific MRSA USA300 was used in this study? There are many USA300 strains.
Thanks for recognizing this. We have added in that this was the CA-MRSA genotype CMRSA10 (USA300)
Lines 212-225: Already the justification why this experiment was performed is way too long (lines 213-222). The methodology was explained on four additional lines (222-225).
Agreed completely and have abbreviated this.
Although mentioned in the method section (lines 128-129), data on viral replication on both cell types or supernatants is not provided by the authors.
Lines 226-253: According to the method section, both cell types were harvested and bacterial numbers were determined (lines 118-120), however, the authors provide data on epithelial cells only. What is the rationale behind that?
-We state in the results that: “A 0.4 μm insert was used, in order to ensure that MRSA was only able to infect alveolar epithelial cells; this mimics what was observed in fatal cases of pH1N1-MRSA co-infection during the 2009 swine flu pandemic”. Also, in the materials and methods (line 121) we state that: “Quantification of the total number of adherent and internalized bacteria was determined at 1, 4, 8, 12, 16, 20, and 24 h post-bacterial infection. Respiratory epithelial HPAEpiCs were harvested for bacterial enumeration by washing 2X with DPBS followed by gentle scraping. Cells were pelleted by centrifugation at 5,000 rpm for 10 min, supernatant removed, and cells resuspended in 0.025% TritonX-100. CFU were quantified by standard bacterial plating on tryptic soy agar (MP Biomedicals, LLC, Solon, OH, USA)”.
Lines 226-253: The author describe nicely what is shown in Fig. 1. However, if there are no significant differences, there is also no need to convince the reader of trends which are not present. The athors conclude correctly (lines 251-252) that there is no impact of H1N1 infection on bacterial replication. Therefore, some of the section can be easily deleted from this results parts. Unfortunately, this kind of arguing continues through the entire result section of the manuscript and should be improved by the authors.
-Agreed completely with the reviewer. We have revised the results to focus solely on those results that were deemed significant.
Figure 1. Is this a representative graph? How many experiments were performed? There are no SDs or SEMs.
SEMs have been added to the graph and we have clarified in the methods that: “Four biological replicates were performed at each time point for enumeration”.
Lines 263-265: The authors state that modulation of analyzed bacterial genes was observed only at 1 and 4 h post infection. Figure 2 depicts only these two time points. No data on other time points is provided.
We appreciate this comment and have added clarification to our statement in the results: “Our prior work with co-infection in A549 cells demonstrated that altered virulence factor expression was only found during early infection (1-4 h post-MRSA addition). Thus, here we focused on the same time points”
Lines 266-267 and lines 270-271: The authors state: each of the 13 genes were upregulated. Again, this is not true. If no significant differences are calculated, consequently no upregulation of the gene is detected.
We have clarified these statements to only list significant observations.
Figure 2: Which statistical test was used?
It was done using ANOVA as is listed in the statistical analysis section of the methods: “Statistical analyses were performed using ANOVA for comparisons of group means using Prism 8 for macOS (version 8.2.1)”.
Figure 3: Is this a representative graph? How many experiments were performed? There are no SDs or SEMs.
We appreciate the reviewer’s comments in regards to the ECIS data and have added the SEMs from our analysis and this included three biological replicates per time point, per condition. This has been added into the Materials and Methods section.
Figure 3 and lines 281-303: the authors describe significant differences, however, the values or "stars" are not provided in the corresponding figure.
We have corrected this.
Lines 281-303: the sentense: "at 90 h, pH1N1-MRSA..." does not make any sence. The data of 90 h is not provided.
This is a great point and appreciate the reviewer mentioning this. Our initial ECIS data was acquired at 45 hours prior to infection with pH1N1 to ensure that the transwells replicate samples were registering on the instrument properly so the initial infection with pH1N1 is listed as time 0 on the graph but came well after our initial quality check. We have modified this accordingly in the text.
I really like the fact that the authors distinguish between the epithelial and capillary compartment in the kinome analyses. However, I mostly disagree with the presented conclusions. To my knowledge, pathway analyses are ranked based on p-values. I addition, the number of significantly differentially expressed proteins out of the total number of proteins which are assigned to a particular pathway are important for such calculations. I will give just one example how I would interprete the data based on Table S2. I would add another row to each table showing how many proteins are assigned to each calculated pathway. Furthermore, I would remove all pathways which have a p-value =0.05 or higher. Indeed, cytokine signalling and signalling by interleukines would remain. But it is not one of the top-ten pathways (based on p-values). I would only be able to claim it, if 56 kinases out of e.g. 200 (totaly assigned) would be covered (=28%) while estrogen signalling would have e.g. 4 out of 200. Just by looking at the data of epithelial cells (Tables S2-S4), I would conclude that e.g. Interferon signalling plays a role in viral infections (Table S3 (last significant pathway);typical viral response) which is dampened by bacterial infection of H1N1 infected cells (Figure S2; p-value>0.05). In bacterial infections itself, this signalling does not play a role (no IFN signaling at all). This is how I would approach it, of course by using the whole data set.
This is a valid point from the reviewer in regards to cutoff p-values for the pathway overrepresentation analysis. We have re-assessed the pathway data and have looked at similar trends across the different infection conditions as well as the associated p-values and uploaded protein counts. We have focused our results on overall trends amongst related pathways within each table.
Similar commets apply to the cytokine data (Figure 6) and the corresponding result section. Please comment only significat differences. Trends are not differences!
Based on the results, I would conclude that EGF does not play a role. In the remaining cytokine analyses, only viral signature is present in co-infections. No bacterial imprint is seen.
Discussion: this part is way too long and needs to be shortened and adjusted to the improved results version.
We appreciated these last few comments and have adjusted the cytokine section to only highlight results that were significant and have substantially shortened the Discussion section.
Reviewer 2 Report
This manuscript is a follow-up study to a previous publication by Nickol et al. (PMID: 30699912). The current study describes a model to mimic the alveolar-capillary barrier using co-culture of primary human alveolar epithelial cells (HPAEpiC) and microvascular endothelial cells (HPMEC). Using this in-vitro model, the impact of prior infection by pandemic A/Mexico/4108/09 (H1N1) on methicillin-resistance staphylococcus aureus (MRSA) secondary infection was studied.
Main findings:
Prior infection with pH1N1/2009 resulted in
- severe dysfunction of the alveolar-capillary barrier caused by MRSA
- modulation of expression of 13 MRSA virulence factors at 1 and 4 h after MRSA infection
- dysregulated host cell signaling (mainly TLR- and inflammatory CM) responses in both epithelial and endothelial cells at the alveolar-capillary barrier
This study adds to our understanding for the molecular mechanisms underlying severe symptoms and high mortality after co-infections of influenza viruses and staphylococcus aureus. The model described herein is useful to study other respiratory viral and bacterial co-infections.
Minor comments
- The GenBank accession number of this Mexican pH1N1 should be added (e.g. ACQ99613, if this the correct sequence) or a reference should be cited (e.g. PMID: 20797971).
- Line114 and elsewhere “Mock-infected with… media” should be changed to just “mock” or “uninfected-mock” or something similar because the mock-cells were not infected, if this right.
- Lines 82 and 112: was trypsin added for the growth of pH1N1? please clarify.
- Line 114: please describe the composition of the “infection media” here or in line 94.
- Line 127: why the RT-qPCR was used to assess the replication of pH1N1?
- Line 133: please change “Viral copy number” to “Viral RNA copy number”.
- Line 212: What about the titer of pH1N1 at the time of MRSA-infection and lateron? This should be written briefly in section 3.1 or should be provided as supplementary data.
- Figure 1: please add the standard deviation/SEM/Error bar to indicate reproducibility of these results and indicate in the legend that the results are the mean/median/? values.
- Figure 2: please describe in the legend under the figure what does the dashed line (cut-off) refer to.
- Figure 2: it is difficult to assign the target gene in the figure legend to the color of the column in the figure. It is recommended to add the gene directly to the x-axis.
- Lines 216-224 can be deleted
- Lines 270-271: this sentence is a redundant to the sentence in lines 268-270
- Line 273: “at 4 h” can be deleted
- Line 397: At 90 h? it is described in line 159 that the integrity was measured for 48 h only. Please clarify/correct.
- Figure 3: please indicate in the legend how many replicates/wells were done/measured and indicate whether the data shown are mean/median of x replicates. If possible, the error bars should be shown in all figures. If the error bars were removed for clarity, please consider adding figures with error bars as supplementary data.
- Figure 6: How to explain the reduction of IP-10 and MCP-1 caused by pH1N1 24 h post-infection with MRSA? Please add interpretation of these results to the discussion section line 654-659.
- Line 480: if MRSA does not affect the secretion of IL6, how to explain the increased IL-6 in pH1N1-MRSA compared to H1N1? This is also in contrast to (1) the discussion section line 641-651 and (2) previous results that SA induced IL-6 expression (e.g. PMID: 7729892; PMID: 21449942 and others cited in this manuscript). pls consider to delete this sentence.
- References: references are not consistent; the title of some references are written in capital/small letters and some journals have abbreviated/full name journals.
- Typos: line 352: “wer” should be “were”
- line 481: “oints” should be “points”
- line 185: should “analyzed” be written “analyze”?
- Figure S1: it would be helpful for the reader to add lines 96-108 to the legend of this figure and indicate the commercial source of cells
- Table S1: please indicate the direction of the (reverse) primers (e.g. 5´- 3´)
- The word “Data” in the manuscript is written sometimes singular or plural, please check and correct if necessary.
In this co-culture model, how does influenza virus support the expression of virulence factors of MRSA without affecting the titer of MRSA? Is that through direct bacteria-virus interaction (e.g. PMID: 31110359) or indirectly via cellular factors triggered by influenza virus (PMID: 24549845) and mentioned on “sbi” line 535.
Author Response
This manuscript is a follow-up study to a previous publication by Nickol et al. (PMID: 30699912). The current study describes a model to mimic the alveolar-capillary barrier using co-culture of primary human alveolar epithelial cells (HPAEpiC) and microvascular endothelial cells (HPMEC). Using this in-vitro model, the impact of prior infection by pandemic A/Mexico/4108/09 (H1N1) on methicillin-resistance staphylococcus aureus (MRSA) secondary infection was studied.
Main findings:
Prior infection with pH1N1/2009 resulted in
severe dysfunction of the alveolar-capillary barrier caused by MRSA
modulation of expression of 13 MRSA virulence factors at 1 and 4 h after MRSA infection
dysregulated host cell signaling (mainly TLR- and inflammatory CM) responses in both epithelial and endothelial cells at the alveolar-capillary barrier
This study adds to our understanding for the molecular mechanisms underlying severe symptoms and high mortality after co-infections of influenza viruses and staphylococcus aureus. The model described herein is useful to study other respiratory viral and bacterial co-infections.
Minor comments
The GenBank accession number of this Mexican pH1N1 should be added (e.g. ACQ99613, if this the correct sequence) or a reference should be cited (e.g. PMID: 20797971).
This has been added.
Line114 and elsewhere “Mock-infected with… media” should be changed to just “mock” or “uninfected-mock” or something similar because the mock-cells were not infected, if this right.
These have been changed.
Lines 82 and 112: was trypsin added for the growth of pH1N1? please clarify.
This has been clarified.
Line 114: please describe the composition of the “infection media” here or in line 94.
This has been clarified on lines 111-114.
Line 127: why the RT-qPCR was used to assess the replication of pH1N1?
Given the small volumes of media that were used to wash the transwells we opted to use RT-qPCR to provide evidence of productive infection as the focus of this manuscript was not on the replication kinetics of pH1N1 during co-infection. We have added a supplemental figure to address viral genome copies post-MRSA addition in the co-infected cells.
Line 133: please change “Viral copy number” to “Viral RNA copy number”.
This has been clarified.
Line 212: What about the titer of pH1N1 at the time of MRSA-infection and lateron? This should be written briefly in section 3.1 or should be provided as supplementary data.
We appreciated this comment and have added a statement on this in Section 3.1 and a supplemental figure (Supp. Fig 1).
Figure 1: please add the standard deviation/SEM/Error bar to indicate reproducibility of these results and indicate in the legend that the results are the mean/median/? values.
These have been added.
Figure 2: please describe in the legend under the figure what does the dashed line (cut-off) refer to.
Thanks. This has been clarified in the legend.
Figure 2: it is difficult to assign the target gene in the figure legend to the color of the column in the figure. It is recommended to add the gene directly to the x-axis.
Completely agreed with the reviewer and the coloring has been changed to make this easier for the reader.
Lines 216-224 can be deleted
This has been addressed.
Lines 270-271: this sentence is a redundant to the sentence in lines 268-270
This section has been reformatted to remove any discussion on trends and the reviewer’s comments have been clarified.
Line 273: “at 4 h” can be deleted
This has been deleted.
Line 397: At 90 h? it is described in line 159 that the integrity was measured for 48 h only. Please clarify/correct.
Agreed with the reviewer. 90hr is the total length of the experiment as there was an initial resting period of the cells alone where resistance measurements were taken to ensure stabilization. We have clarified our statements in the manuscript to reflect only the time from the point of influenza virus infection onwards.
Figure 3: please indicate in the legend how many replicates/wells were done/measured and indicate whether the data shown are mean/median of x replicates. If possible, the error bars should be shown in all figures. If the error bars were removed for clarity, please consider adding figures with error bars as supplementary data.
This has been clarified in the Materials and Methods and error bars have been added to the figure to reflect any variations found between biological replicates.
Figure 6: How to explain the reduction of IP-10 and MCP-1 caused by pH1N1 24 h post-infection with MRSA? Please add interpretation of these results to the discussion section line 654-659.
We have added discussion on these observations as well as additional references to support these data points.
Line 480: if MRSA does not affect the secretion of IL6, how to explain the increased IL-6 in pH1N1-MRSA compared to H1N1? This is also in contrast to (1) the discussion section line 641-651 and (2) previous results that SA induced IL-6 expression (e.g. PMID: 7729892; PMID: 21449942 and others cited in this manuscript). pls consider to delete this sentence.
We have deleted the sentence in line 480 from the original submission.
References: references are not consistent; the title of some references are written in capital/small letters and some journals have abbreviated/full name journals.
We have ensured that the Endnote style used for the references are done according to MDPI style as required by the journal.
Typos: line 352: “wer” should be “were”
line 481: “oints” should be “points”
line 185: should “analyzed” be written “analyze”?
All of these have been corrected.
Figure S1: it would be helpful for the reader to add lines 96-108 to the legend of this figure and indicate the commercial source of cells
We have amended the figure legend for Fig. S1.
Table S1: please indicate the direction of the (reverse) primers (e.g. 5´- 3´)
This has been added.
The word “Data” in the manuscript is written sometimes singular or plural, please check and correct if necessary.
This has been corrected.
In this co-culture model, how does influenza virus support the expression of virulence factors of MRSA without affecting the titer of MRSA? Is that through direct bacteria-virus interaction (e.g. PMID: 31110359) or indirectly via cellular factors triggered by influenza virus (PMID: 24549845) and mentioned on “sbi” line 535.
This is a great question. We have added a point of clarification to this in the Discussion but our hypothesis is that this is due to interaction with host factors (secreted or exposed cell receptors following pH1N1 infection) that lead to activation of two-component signaling systems or sensory systems and resulting in augmentation of an adhesion and invasion phenotype.
Round 2
Reviewer 1 Report
The authors tried to improve the quality of the manuscript. However, it is hard to see the changes, particularly after the conversition of the word file with track changes to pdf. Especially, The figure legends for Figure 1 and 3 are still incomplete.
Specific commetns:
- Although the authors improved the method section, certain information, inlucluding (i) n=x, (ii) which statistical test was used etc. should be provided in the figure legends. This makes it easier to understand and judge the result.
- lines 264-265, Figure S2: Why are the viral copy numbers dropping? Was a control experiment without bacterial infection performed? What might be the reason for the drop? Cytotoxicity might be one explanation. Was this quantified by the authors?
- Refering to the previous comments: Although mentioned in the method section (lines 128-129), data on viral replication on both cell types or supernatants is not provided by the authors. AND Lines 226-253: According to the method section, both cell types were harvested and bacterial numbers were determined (lines 118-120), however, the authors provide data on epithelial cells only. What is the rationale behind that? The explanation by the authors is not sufficient. I agree that the bacteria might not pass the 0.4 µm filter. However, the virus does. Was the viral replication verified in both cell types?
- I am also a bit confused by the correction of the previously commented supplementary tables. I have to admit that I am not an expert in kinome analyses. However, based on the presented kinome data, I would assume that all kinases were found in the analyses = 100% match, which is hard to believe. I will give just a few examples. Mitotic G2-G2/M phases: according to the authors 10 uploaded kinases out of 10 assigned kinases were found in the analyses (Fig. S2). Are there really only 10 kinases assigned to this pathway. Particularly, I started to wonder about the cytokine signalling pathway. 56 out of 56 kinases were found by the authors in the analyses (100% match). Particularly in cytokine signaling, which comprises of all cytokine processes, I would assume that more than 56 kinases are involved. Each receptor and subsequent regulator would need a kinase. According to the reactome data: 1332 molecules are assigned to this pathwas. 981/1332 are proteins. Are there really only 56 kinases in this pathway? And how precise is the analysis, which is performed by the authors? I have never seen a 100% coverage before.
Author Response
Responses to Reviewers:
- Although the authors improved the method section, certain information, inlucluding (i) n=x, (ii) which statistical test was used etc. should be provided in the figure legends. This makes it easier to understand and judge the result.
These have been added.
- lines 264-265, Figure S2: Why are the viral copy numbers dropping? Was a control experiment without bacterial infection performed? What might be the reason for the drop? Cytotoxicity might be one explanation. Was this quantified by the authors?
This is a great question. Viral loads were taken from the co-infected samples alone as these were the only samples where resistance decreased over time through our ECIS analysis. Given the decrease in resistance reminiscent of our prior analysis in A549 cells, the decreasing viral loads are most likely reflective of decreasing cell viability. We did not take samples to assess this by LDH as the cells are grown at the air-liquid interface and secreted factors in the supernatant are reflective of both the epithelial and endothelial cells. Western blot analysis of ZO-1 and ZO-2 suggested that tight junction protein expression was also decreased in the alveolar epithelial cells as compared to either condition providing additional evidence for decreased cell viability.
- Refering to the previous comments: Although mentioned in the method section (lines 128-129), data on viral replication on both cell types or supernatants is not provided by the authors. AND Lines 226-253: According to the method section, both cell types were harvested and bacterial numbers were determined (lines 118-120), however, the authors provide data on epithelial cells only. What is the rationale behind that? The explanation by the authors is not sufficient. I agree that the bacteria might not pass the 0.4 µm filter. However, the virus does. Was the viral replication verified in both cell types?
We did RT-qPCR for virus on the endothelial cells from our transwells and no signal was identified on multiple samples from the endothelial cells although the epithelial cells were positive. For the bacterial enumeration, bacterial plating on endothelial cells rendered no colonies so we confirmed the filtering by adding MRSA alone on one side of a 0.4 µm filter with MH broth in the well and incubated for 24 hours. Direct addition of MRSA to media in the well resulted in a turbid solution following incubation while the media in the wells containing transwells + MRSA had no bacterial growth. We have also clarified that no bacteria or virus was identified in the endothelial cells in Section 3.1.
- I am also a bit confused by the correction of the previously commented supplementary tables. I have to admit that I am not an expert in kinome analyses. However, based on the presented kinome data, I would assume that all kinases were found in the analyses = 100% match, which is hard to believe. I will give just a few examples. Mitotic G2-G2/M phases: according to the authors 10 uploaded kinases out of 10 assigned kinases were found in the analyses (Fig. S2). Are there really only 10 kinases assigned to this pathway. Particularly, I started to wonder about the cytokine signalling pathway. 56 out of 56 kinases were found by the authors in the analyses (100% match). Particularly in cytokine signaling, which comprises of all cytokine processes, I would assume that more than 56 kinases are involved. Each receptor and subsequent regulator would need a kinase. According to the reactome data: 1332 molecules are assigned to this pathwas. 981/1332 are proteins. Are there really only 56 kinases in this pathway? And how precise is the analysis, which is performed by the authors? I have never seen a 100% coverage before.
We apologize for the confusion on this table and the subsequent analysis of the kinome array data through InnateDB. We have added additional columns for clarification including those proteins where expression and p-values were above the threshold values and all of the gene symbols for proteins that were identified in our data and fit into the pathways as identified by InnateDB. InnateDB utilizes the entire data (including up- and down-regulated phosphorylation events) set to look at the overall trends of activation (upregulation) and associated p-values in all genes found within the identified pathways of interest. Based on the trends and threshold boundaries set by the user (FC>1.5; p<0.05 for our data), InnateDB will identify biological responses that are overrepresented within the complete data sets based on an overall p-value.